# Low-Density sEMG-Based Pattern Recognition of Unrelated Movements Rejection for Wrist Joint Rehabilitation

**DOI:** 10.3390/mi14030555

**Published:** 2023-02-27

**Authors:** Dongdong Bu, Shuxiang Guo, Jin Guo, He Li, Hanze Wang

**Affiliations:** 1School of Life Science, Beijing Institute of Technology, Beijing 100081, China; 2Key Laboratory of Convergence Medical Engineering System and Healthcare Technology, Ministry of Industry and Information Technology, School of Life Science, Beijing Institute of Technology, Beijing 100081, China

**Keywords:** surface electromyography (sEMG), wrist joint rehabilitation training, unrelated movements rejection, convolutional neural network (CNN), autoencoder (AE)

## Abstract

sEMG-based pattern recognition commonly assumes a limited number of target categories, and the classifiers often predict each target category depending on probability. In wrist rehabilitation training, the patients may make movements that do not belong to the target category unconsciously. However, most pattern recognition methods can only identify limited patterns and are prone to be disturbed by abnormal movement, especially for wrist joint movements. To address the above the problem, a sEMG-based rejection method for unrelated movements is proposed to identify wrist joint unrelated movements using center loss. In this paper, the sEMG signal collected by the Myo armband is used as the input of the sEMG control method. First, the sEMG signal is processed by sliding signal window and image coding. Then, the CNN with center loss and softmax loss is used to describe the spatial information from the sEMG image to extract discriminative features and target movement recognition. Finally, the deep spatial information is used to train the AE to reject unrelated movements based on the reconstruction loss. The results show that the proposed method can realize the target movements recognition and reject unrelated movements with an F-score of 93.4% and a rejection accuracy of 95% when the recall is 0.9, which reveals the effectiveness of the proposed method.

## 1. Introduction

The loss of one side of upper limb mobility function and sensory information hampers the activities of daily living (ADL) for hemiplegic patients [1]. Many clinical studies have shown that the plasticity of the central nervous system can restore the function of the injured central nervous system. This remodeling can be strengthened and consolidated through continuous motor relearning to help the functional repair and reconstruction of the neuromuscular system. Therefore, appropriate rehabilitation training can help patients recover their motor function to a certain extent. The delay of intervention therapy will affect patients’ affected side functional capability restoration and increase rehabilitation duration [2,3]. Bilateral rehabilitation training is considered to be an effective strategy for the rehabilitation of hemiplegic patients [4]. The maximum latency of 300ms was recommended for sEMG-based pattern recognition method [5]. The sEMG-based pattern recognition method mainly finds the internal information of multi-channel sEMG signals through feature learning, which assumes that the internal information of sEMG signals for the same movements are similar, but the internal information of sEMG signals for different movements are different [6]. Therefore, the upper limb intention recognition with high accuracy and reliability is an important part of the exoskeleton control system [7], especially for remote monitoring which needs a more efficient and intelligent framework to decode the subject’s intention. sEMG signals can directly reflect the activation of superficial muscles with rich limp motion control information. In recent years, sEMG signal as the exoskeleton control signal source has been widely applied to a rehabilitation estimation [8,9,10], human intention prediction [11,12,13,14,15,16,17], and rehabilitation robot control [18,19,20,21,22,23].

However, in practical rehabilitation training, there are still great limitations in clinical application, such as multi-user difference, electrode offset, unrelated movements interference, and muscle crosstalk, which are important factors restricting the practicability of sEMG pattern recognition methods. Many studies reveal that deep learning with CNN has great advantages in mining information [24,25]. For the interference of the unrelated movement, sEMG is very sensitive to muscle contraction, and slight changes in upper limb movements may cause interference with the exoskeleton control system. Because the signals of unrelated movements are not trained by the classifier, the classifier will inevitably make wrong decisions. Therefore, these unrelated movements will cause the system to provide incorrect instructions, which may reduce the user experience and also cause security accidents. In recent years, many researchers have studied this problem, and the existing solutions can be roughly divided into three methods: domain-based methods, probability-based methods, and reconstruction-based methods. These studies mainly devise a filtering method that can only identify target samples and remove unrelated samples. Q. Ding et.al [26]. proposed methods to reduce or eliminate the impacts of three types of daily interferences on myoelectric pattern recognition, that is, outlier motion, muscle fatigue, and electrode doffing/donning to reveal the potential of the adaptive incremental hybrid classifier for myoelectric pattern recognition strategy in the development of clinical myoelectric prostheses. J.W. Robertson et.al. [27] found that confidence-based rejection improves usability outcomes for support vector machine-driven myoelectric control. However, the above research on the rejection of unrelated samples often assumes that the distribution of unrelated samples and target samples is significantly different, so a relatively simple method can be used to distinguish it.

The unrelated movements are unpredictable and often very similar to the target movements, especially for wrist rehabilitation training. Therefore, the rejection of wrist-joint-unrelated movements is of interest to this paper. L. Wu et al. [28] developed a robust myoelectric control method for rejecting novel/unknown patterns based on high-density surface electromyogram (HD-sEMG) signals, which enhanced the robustness of the myoelectric pattern recognition systems. There are many small and concentrated muscle groups in the forearm, so the sEMG signal is greatly affected by muscle crosstalk. Because of the synergetic effect of these muscles, some small unrelated movements will greatly affect the accuracy of wrist motion recognition. In addition, considering the requirements of remote rehabilitation, it is necessary to combine sEMG acquisition equipment with upper limb exoskeleton equipment to form a portable upper limb rehabilitation system. To mine available information differences between target movement samples and unrelated movement samples and meet the requirements of a remote wearable exoskeleton rehabilitation system, the main problem of this paper is to solve the wrist joint unrelated movements interference and improve the robustness of wrist joint movement recognition. In this paper, aiming at low-density sEMG signals, a convolutional neural network (CNN) with autoencoder (AE) hybrid neural network structure with softmax loss and center loss (CNN-AE-SC) is proposed to achieve upper limb wrist movements classification and unrelated movements rejection. The hybrid network consists of two parts: the CNN network and the AE network. The CNN with central loss and softmax loss is trained to extract the deep features of the sEMG signal, whereas the AE network is applied to obtain the reconstructed loss to reject unrelated movements, thus realizing accurate rejection of dynamic and complex unrelated movements.

The rest of the paper is organized as follows. In Section 2, the experimental protocol and methods are described, which mainly include signal processing and image encoding, discriminative features extraction, unrelated movements rejection module, and evaluation criteria. In Section 2, the results and discussion of the study are reported. Finally, the conclusion is presented in Section 4.

## 2. Methods

### 2.1. System Overview

Bilateral upper limb rehabilitation is a rehabilitation training strategy in which the healthy side drives the affected side to perform synchronous movements [4]. In a previous study by our research group, a gear-driven powered exoskeleton device was developed [8,9,10]. As shown in Figure 1, the sEMG signals for different upper limb movements are collected. After signal preprocessing and feature extraction, the features are input into the trained model to achieve pattern recognition. The recognition results are used to control the upper limb exoskeleton worn on the affected limb. Finally, the exoskeleton assists the upper limb in performing desired movements according to the motion intention of the subjects.

Accurate and reliable motion intention perception and prediction is the key to the exoskeleton control system, especially for remote monitoring. In this paper, the rejection of unrelated movements is considered to achieve rehabilitation training with high accuracy and reliability. The proposed method overview is shown in Figure 1b, which mainly includes sEMG image encoding, discriminative features extract module, and unrelated movements rejection module. The method proposed in this paper is based on the assumption that although the signals between the wrist joint target movements and unrelated movements are similar, these movements are activated by different muscle combinations. Therefore, the deep spatial features of the sEMG images extracted from these movements are also different. First, the sEMG signal is processed by a sliding signal window and image coding, and then CNN is used to extract deep discrimination features from these sEMG images. CNN is also used to predict the possible target category of the sample combined with the center loss and softmax with cross-entropy loss, and then the AE network is trained to reject samples higher than the set threshold by calculating reconstruction error. If the reconstruction error is below a threshold, it is considered that the sample belongs to the target movements, and the movements category is the prediction result of CNN; If the reconstruction error is above the threshold, the sample is considered as unrelated movements and rejected. These steps are introduced in detail below.

### 2.2. sEMG Dataset

The sEMG data contains two distinct sub-datasets in this paper, which are The NinaPro DB5 [29] dataset and NewMyo dataset. The NinaPro DB5 dataset, recorded with the Myo Armband, contains data from 10 able-bodied participants performing a total of 53 different movements (including neutral which denotes resting state) divided into three exercise sets. The NewMyo dataset is collected from two able-bodied subjects in this paper (one male and one female; right-hand; age from 22 to 32). The experimental content was introduced to each subject in detail, and each subject signed the informed consent. All the experimental procedures are approved by the Institutional Review Board (IRB) in the faculty of Engineering Kagawa University (Ref. No. 01-011 from February 2020), which follows the ethical principle of declaration of Helsinki.

The experimental scheme adopted in this paper is shown in Figure 2. sEMG signals are collected using the Myo armband (Thalmic Labs Inc. Kitchener, Ontario, Canada), which is a commercially available device with eight equidistance sEMG sensors that transfers the data through a Bluetooth low-energy connection to the computer. The Myo armband sensors distribution is shown in Figure 2a, in which the electrode with the LED light and Myo logo that shows the sync state is channel 4, followed by channel 5 in a clockwise direction and channel 3 in a counterclockwise direction. Limited by the number of Myo armbands, the red dotted line frame is the main muscle group of wrist joint motion, including brachioradialis, flexor carpi radialis, extensor carpi radialis, palmaris longus, flexor carpi ulnaris, and so on. The initial position of channel 4 is placed about 2 cm away from the right elbow joint, which is located in the brachioradialis muscle. The real-time sEMG data at 200 Hz can be acquired through the software development kit (SDK) of the Myo armband. The Myo armband includes a 50 Hz notch filter which eliminates the power-line interference (50 Hz) [30]. The subjects were asked to perform 12 different movements, including seven target movements and five rejection movements which are highly similar to the sample signal of the target movements, as shown in Figure 2b,c. Seven target movements include wrist pronation (T1)/supination (T2) (axis: little finger), wrist extension (T3)/flexion (T4), wrist radial deviation (T5)/ulnar deviation (T6), and neutral. The five rejection movements include the abduction of all fingers (R1), fingers flexed together in a fist (R2), wrist pronation (R3)/supination (R4) (axis: middle finger), and wrist extension with closed hand (R5). The data collection scheme strictly replicated the scheme from Ninapro DB5. The subjects were asked to perform the movements with the right hand for six repetitions. Each movement repetition lasted 5 s and was followed by 3 s of rest to avoid muscular and mental fatigue of the subjects.

For the NinaPro DB5 dataset, the second exercise set is of particular interest to this paper. The first eight-channel sEMG signals of seven target movement signals and five unrelated movement signals related to this study are obtained as shown in Figure 2b,c. The purpose of this paper is to achieve wrist target movements classification and unrelated movements rejection based on the low-density sEMG signals. Considering the robustness of the model, the sEMG signals of each target motion for 12 recruiters are randomly sampled in the proportion of 60%, 20%, and 20% to form the training dataset, verification dataset, and test dataset depending on target motion categories. The training dataset is used to train the model. The validation dataset is used to calculate the average reconstruction error of target movement samples. The test data consists of the test dataset and all unrelated movement samples, which are used to evaluate the performance of the model proposed in this paper.

### 2.3. Data Segmentation and sEMG Image Encoding

The sEMG signal is collected through the Myo armband in this paper, which can be regarded as a low-density array signal, including additional spatial information. CNN is good at extracting spatial information. Therefore, this procedure preprocesses low-density sEMG signals into images. First, the sliding windows are used to segment the multi-channel sEMG signals into a series of overlapped analysis windows. Limited by real-time requirements, the maximum latency of 300 ms was recommended in [31]. Therefore, the window length and window step are set as 250 ms and 100 ms, respectively. Therefore, sEMG is segmented into a series of sEMG window matrices of size 50×8. Then, the amplitude mean and standard deviation of all the signals in the neutral state are calculated, and the amplitude threshold method is used to filter each signal window signal to obtain the active segment signal of the wrist joint. The threshold is defined as the sum of the mean and three times the standard deviation of the neutral signal, as shown in the Equation (1).
(1)$$\mathrm{Threshold}={\;\mu }_{\mathrm{neutral}}+3\sigma_{\mathrm{neutral}}$$
where Threshold is the boundary threshold between the active segment and the neutral segment, which is used to obtain signals of the muscle active segment. If the amplitude of a window signal is above the Threshold, it can be considered as a muscle activity state. The $\mu_{\mathrm{static}}$ and $\sigma_{\mathrm{static}}$ denote the expectation and standard deviation of all neutral signals.

Then three time-domain (TD) features are extracted for the signals of muscle activity state, that is, root mean square (RMS), mean absolute value (MAV), and waveform length (WL). They are calculated with Equations (2)–(4). The RMS, MAV, and WL are the most commonly used TD features in motion pattern recognition [5,7,10,12,30], and they are closely associated with the amplitude of sEMG, which can directly reflect the intensity of muscle contraction with low computational complexity. The studies [7,28,32] also showed that the combination of TD features and a small-scale CNN could reach satisfactory control performance. Therefore, the CNN combined with three time-domain features is applied in this paper with the advantages of simple network structure, less computation, and excellent results.
(2)$$\mathrm{RMS}=\sqrt{\frac{1}{N}\sum_{n=1}^{N}x_{n}^{2}}$$
(3)$$\mathrm{MAV}=\frac{1}{N}\sum_{n=1}^{N}\left|x_{n}\right|$$
(4)$$\;\mathrm{WL}=\sum_{n=2}^{N}\left|x_{n}-x_{n-1}\right|$$
where $x_{n}$ denotes the value of the $n_{\mathrm{th}}$ point for each window signal. Therefore, each window signal belonging to muscle activity is transformed into a 3×53×8 feature matrix as shown in Figure 3, which can directly represent the state of muscle activity.

### 2.4. The Discriminative Feature Extraction Network

Many studies [6,15,24,25,26,27,28,29,30] have verified the effectiveness of the convolution neural network for mining the internal information of sEMG signal. In this procedure, CNN is used to extract discriminative features from low-density sEMG images. The input of CNN is the sEMG feature images. This module makes two contributions: one is to provide AE with the deep discriminative features of sEMG signals, and the other is to identify target movements. The architecture of the CNN developed in this paper is shown in Figure 4. This CNN contains six layers, including three convolutional layers, two fully-connected layers, and an output layer. Three 2D convolution kernels (16×2 filter with a stride of 2 and a padding of 0, 8×3 filter with a stride of 2,1 and a padding of 2,1, and 3×3 filter with a stride of 1 and a padding of 1,1) are used to process the sEMG images for the first three convolution layers in turn, and the convolution layer activation function is LeakyReLU activation. During training, the dropout rate between Layer4 and Layer5 is set to 20% to prevent model overfitting. The adaptive moment estimation (Adam) algorithm is adopted with the batch size set to 32 to train the network. We optimized the network using the training samples for 200 epochs with a learning rate of 0.0001. The trained CNN is used for extracting discriminant features, and the parameters are fixed during the training of the AE. The center loss and softmax loss are applied to optimize the network, which makes the deep characteristics of the fifth layer output of the network discriminative. Generally, only softmax loss is used for CNN model training, as shown in Equation (5).
(5) Lsoftmax=−1N∑i=1Nlogexpxclass∑mexpxm=−1N∑i=1NlogewyiTxi+byi∑m=1MewmTxi+bm
where xi∈ℝd, and yi denote deep features and label of the ith record, respectively. wm is the mth column of the parameters W=w1,w2…,wm∈ℝd×m. b∈ℝc is the bias. N and M denote the length of mini-batch samples and the number of categories, respectively. The softmax with cross-entropy loss is good at learning inter-class information, which describes the distance of two probability distributions [33]. For the multi-classification task, softmax activation can map the deep features to a category probability distribution, and cross-entropy can describe the loss between real label value and predicted probability. The lower the softmax with cross-entropy loss, the better separability of deep features. However, the CNN trained only with softmax loss (CNN-AE-S) is more interested in the accuracy of the prediction probability for correct labels while ignoring the differences of other incorrect labels, resulting in the scattered distribution of learned features. That is, the optimization of intra-class distance is relatively weak, resulting in the poor discriminant performance of the model. In general, especially for a very similar sample distribution, small intra-class differences and large inter-class differences need to be considered for model training. Therefore, inspired by A. Farzaneh et al. [33], the center loss combined with the softmax loss is used to train the network. According to [33], the center loss can be calculated as shown in Equations (6)–(10).
(6)Lcenter=−12N∑i=1N‖xi−cyi‖22
where cyi∈ℝd is the yith class center of deep features. cyi will be updated with the change of deep features. Therefore, it will be inefficient and even impractical to average the features of each class in each iteration for all training datasets. Therefore, in this paper, cyi is updated using a mini-batch sample for each iteration. And considering mislabeling samples caused by large disturbances, the scalars α (restricted in 0,1) are used to control the learning rate of center cyi. Therefore, the gradients of Lcenter for xi and the update of cyi are defined as:(7)∂Lcenter∂xi= xi−cyi
(8)Δcj=∑i=1Nδyi=j⋅cj−xi1+∑i=1Nδyi=j
(9)cjt+1= cjt−α⋅Δcjt
When yi is different from j of cj (that is, δcondition=0), cj will not be updated. Conversely, δcondition=1 if yi is the same as j of cj, cj will be updated. Therefore, the loss function in this paper is defined as weighted the softmax loss and center loss as shown in Equation (10). The scalar λ is used to balance two loss functions. The softmax loss is used to increase inter-class spacing, and center loss is used to reduce intra-class spacing, thus making the deep features more discriminative.
(10)Loss= Lsoftmax+λLcenter=−1N∑i=1NlogewyiTxi+byi∑m=1MewmTxi+bm+λ2N∑i=1N‖xi−cyi‖22

### 2.5. Unrelated Movements Rejection Module

In this part, the AE network is trained to reject any samples that do not belong to the target movements by calculating reconstruction error. The CNN model developed in this paper can recognize target movements well, but it does not recognize the performance to reject unrelated movements. The AE network framework adopted by the proposed method is shown in Figure 5. The network adopts a mirror architecture, which can be divided into an encoder and a decoder. The encoder compresses the input into a low dimensional feature space, which only retains the main mode of the input data, and then the decoder can recover the raw data. If the distribution of the input and training data is different, then there are obvious differences in the output of the encoder, and the decoder is unable to recover the input data causing large reconstruction errors. The mean square error (MSE) loss is used to train AE. The input of AE is the layer5 output (512×1) of the CNN network on the training dataset of the target movement samples. The Adam algorithm with a mini-batch (set to 32) is adopted to optimize the parameters in the network. The learning rate is set as 0.0001 with the 1000 epochs to train the AE network. To test the discriminant network, the Pearson correlation (PC) [16] is applied to measure the difference between the input data and the reconstructed data as shown in Equation (11).
(11)$$\rho \left(x_{i},y_{i}\right)=\frac{\mathrm{cov}\left(x_{i},y_{i}\right)}{\sigma \left(x_{i}\right)\cdot \sigma \left(y_{i}\right)}=\frac{\sum_{j=1}^{J}\left(x_{\mathrm{ij}}-{\bar{x}}_{\mathrm{ij}}\right)\left(y_{\mathrm{ij}}-{\bar{y}}_{\mathrm{ij}}\right)}{\sqrt{\sum_{j=1}^{J}{\left(x_{\mathrm{ij}}-{\bar{x}}_{\mathrm{ij}}\right)}^{2}}\sqrt{\sum_{j=1}^{J}{\left(y_{\mathrm{ij}}-{\bar{y}}_{\mathrm{ij}}\right)}^{2}}}$$
where $x_{\mathrm{ij}}$ and $y_{\mathrm{ij}}$ represent the $j_{\mathrm{th}}$ dimension values of the $i_{\mathrm{th}}$ input vector $x_{i}$ and the reconstructed vector $y_{i}$, respectively, and ${\bar{x}}_{\mathrm{ij}}$ and ${\bar{y}}_{\mathrm{ij}}$ denote average of $x_{i}$ and $y_{i}$, respectively. This distance can normalize the high-dimensional vector and calculate the difference. When an unknown sample is input into the model trained in this paper, the discriminative features of the sample are extracted through the CNN, and then these features are input into the trained AE network to calculate the reconstruction error. If the reconstruction error is below the threshold, it is considered that the sample belongs to the target movements, and the motion category is predicted by the CNN. If the reconstruction error is above the threshold, the sample is considered as unrelated movements and rejected. Therefore, it is necessary to select an appropriate reconstruction error threshold to distinguish between target movements and denial movements. Therefore, a prior recall factor is defined, and when the mean reconstruction error on the verification dataset of the target samples reaches the prior recall, this reconstruction error is recorded. Finally, this value is defined as the threshold to distinguish between movements and unrelated movements.

### 2.6. Evaluation Criteria

In this paper, accuracy (Acc) [5], F-score, and receiver operating characteristic (ROC) curve are used to verify the performance of the proposed method. The corresponding equations are defined as follows.
(12)$$\mathrm{Acc}=\frac{\mathrm{TP}+\mathrm{TN}}{\mathrm{TP}+\mathrm{FP}+\mathrm{TN}+\mathrm{FN}}\times 100\%$$
(13)$${\mathrm{Precision}}_{j}=\frac{{\mathrm{TP}}_{i}}{{\mathrm{TP}}_{i}+{\mathrm{FP}}_{i}}\times 100\%$$
(14)$${\mathrm{TPR}}_{j}={\mathrm{Recall}}_{j}=\frac{{\mathrm{TP}}_{i}}{{\mathrm{TP}}_{i}+{\mathrm{FN}}_{i}}\times 100\%$$
(15)$${\mathrm{FPR}}_{j}=\frac{{\mathrm{FP}}_{i}}{{\mathrm{TN}}_{i}+{\mathrm{FP}}_{i}}\times 100\%$$
(16)$$F-\mathrm{score}\;=\frac{2\times {\mathrm{Recall}}_{j}\times {\mathrm{Precision}}_{j}}{{\mathrm{Recall}}_{j}+{\mathrm{Precision}}_{j}}\times 100\%$$
where the subscript j represents the index of different movements, j =1, 2, …, 7 correspond to the T1–T6 and neural movements, respectively. As shown in Equations (12)–(16), TP represents true positive, FP represents false positive, TN represents true negative, and FN represents false negative. The curve (ROC) indicates that the model is better at distinguishing unrelated samples. It is a binary classification problem to judge the target samples and unrelated samples. False positive rate (FPR) represents the probability of the target movement samples (FP) accounting for among all the samples whose prediction results are unrelated movement samples; true positive rate (TPR) represents the probability of the target movement samples predicted by model accounts for all target movement samples. When TPR is 1 and FPR is 0, the accuracy of the model for secondary classification reaches 100%. The area under the curve (AUC) further quantifies the performance of the model represented by the ROC curve. The AUC value is in the range of 0–1. Therefore, the larger the AUC value, the higher the overall performance of the model.

## 3. Experiment Results and Discussion

In this section, the model obtained through the proposed method is presented and discussed. To verify the effectiveness of center loss, the visualization of deep features is provided on the training dataset and we also discussed the influence of hyperparameter λ on the model. To verify the performance of the model, the reconstruction error distribution on the validation dataset and the reconstruction error distribution of all unrelated movement samples are provided. Different evaluation indicators are adopted for quantitative evaluation to compare the performance of different methods.

### 3.1. Influence of Different Hyper-Parameter λ

To verify that the center loss function is available for optimizing deep features to make them separable and discriminative, the t-distributed stochastic neighbor embedding (t-SNE) [34] is applied to reveal the distributions of the deep features with different λ values, as shown in Figure 6a–i. Figure 6a shows the distributions of deep features trained only with the softmax loss using the t-SNE visualization method. Although these deep features have good separability in high-dimensions, the intra-class cluster capability of samples is inadequate and even overlaps. Therefore, these features have poor discrimination and may be unavailable and difficult for the rejection of unrelated movements. As shown in Figure 6b–i, the two-dimensional deep features distribution of training samples is shown when λ is $5\times 10^{-6}$, $1\times 10^{-5}$, $1.5\times 10^{-5}$, $2\times 10^{-5}$, $1\times 10^{-4}$, $2.5\times 10^{-4}$, $5.0\times 10^{-4}$, and $1\times 10^{-3}$, respectively. The hyperparameter λ impacts the model performance by influencing the distributions of the deep features. It can be seen that the larger the λ value, the smaller the intra-class spacing, that is, the more compact the deep features of each category are extracted. The joint supervision loss is used to train CNN, which reduces the inter-class spacing of deep features for samples, and the intra-class spacing is also effectively reduced simultaneously, which shows the effectiveness of center loss for the distribution of deep features where λ denotes the weight of center loss. Figure 6a–i shows that the deep feature distribution is affected by the hyperparameter λ, which is crucial to the rejection performance of unrelated movements.

As shown in Figure 7, the mean F-score and mean accuracy of all subjects of the training model are calculated with λ value defined as $5\times 10^{-6}$, $1\times 10^{-5}$, $1.5\times 10^{-5}$, $2\times 10^{-5}$, $1\times 10^{-4}$, $2.5\times 10^{-4}$, $5.0\times 10^{-4}$, and $1\times 10^{-3}$, respectively. As shown in Figure 7a,b, the green curve indicates the results of the mean f-score and accuracy when the recall is defined as 0.9, and the blue curve indicates the experimental results of the mean f-score and accuracy when the recall is defined as 0.85. When λ=0, only the softmax loss is considered to train the model. The mean F-score and mean accuracy are 0.825 and 0.822, corresponding (recall = 0.9) with the worst performance. When λ>0 (considering the center loss), the model performance is improved, and the model performance will change according to the value of λ. In addition, the increase of λ improves the rejection performance of the model, but the mean f-score and accuracy are reduced with the increase of λ. This is due to the large λ that may greatly compress the intra-class spacing, and CNN is not sensitive to the small differences between samples. Therefore, an appropriate λ is important for deep feature extraction and rejection of unrelated motions, which should be defined according to the practical application. In this paper, λ is defined as 0.00025, which shows better the performance of the model.

### 3.2. Performance of the Trained Model

In this section, the threshold is used to distinguish the target movements from the unrelated movements, and the threshold is determined according to the reconstruction error. Specifically, first a recall factor value for all target movements is preset, then the numerical value is recorded when the reconstruction error of all target movement classes in the verification set reaches the recall rate, and this value is regarded as the rejection threshold of each class. The ROC curve is used to estimate the performance of the proposed method for the target movements and rejection movements under different thresholds. Furthermore, the area under the curve (AUC) further quantifies the performance of the model represented by the ROC curve. The larger the AUC value, the higher the overall performance of the classification model. Then, the average recognition accuracy and F-score of the trained model are calculated in the test dataset and unrelated samples depending on the preset recall factor. The comparison methods are performed: Support vector machines(SVM), linear discriminant analysis (LDA), and LDA with Mahalanobis distance (LDA-MD), CNN-AE-S, CNN-AE-SC.

A larger recall factor will make the threshold of all target movements categories larger, and more reconstruction errors in target movements samples are lower than the threshold. The smaller recall factor will reduce the threshold value of the target movements category, inevitably reducing the target movements recognition performance, but it will strengthen the rejection performance of unrelated movements. The reconstruction error of the target motion is relatively small, whereas the reconstruction error of the unrelated motion samples is significantly higher than that of the target category samples. As shown in Figure 8a, the reconstruction loss distribution is described of target movements on the validation dataset, and Figure 8b shows the reconstruction loss distribution of all unrelated movements in the test dataset by the trained model (λ=0.00025). When the recall of the validation dataset achieves 0.80, 0.85, and 0.90, the corresponding thresholds are obtained, which are 0.0170, 0.0194, and 0.0235, respectively. As shown in Figure 8a,b, it is apparent that most of the sample reconstruction errors in the verification dataset are distributed on the left side of the threshold, whereas most of the reconstruction errors of unrelated samples are distributed on the right side of the threshold. Therefore, there is a threshold to distinguish target movements and unrelated movements.

Figure 8c,d show the PC distribution for each movement, that is, the reconstruction error distribution of each category sample after the trained model on the test dataset when the recall of the model is 0.85 and 0.9, respectively. The blue dotted line indicates the rejection performance of the unrelated movements, which are 98.3% and 95.0%. Therefore, an appropriate threshold can effectively segment the target and unrelated samples.

As shown in Figure 9, The true positive rate (TPR) and false positive rate (FPR) are obtained under different thresholds, and their ROC curves are described. The proposed method reaches the maximum AUC value of 94.3%, which is- higher than that from the CNN-AE with softmax loss (the AUC value of 92.1%) when the recall is 0.9. Therefore, there is a threshold that can obtain the highest TPR while keeping the FPR very low, making the target movements more distinguishable from unrelated movements. In addition, the discriminative deep features combined with simple AE can achieve rejection performance. According to Figure 6 and Figure 8, the center loss can compact intra-class variations and enlarge inter-class differences to improve rejection performance.

Figure 10 shows the confusion matrices by the proposed method (λ = 0.00025) and CNN-AE-S (λ = 0) when the recall factor is defined as 0.85 and 0.9, respectively. The confusion matrices averaged over all subjects with the proposed method on the test dataset are shown in Figure 10a,b. When the recall is 0.9, the mean classification accuracy is 92% and the rejection performance of unrelated motions reached 95% by the CNN-AE-SC method, which is higher than that (82.2% and 71.6%) from the CNN-AE-S. When the recall is 0.85, the mean classification accuracy is 92% and the rejection performance of unrelated motions reached 95% by the CNN-AE-SC method, which is higher than that (82.2% and 71.6%) from the CNN-AE-S. According to Figure 10, for the model trained with the CNN-AE-SC method, all misclassifications of target motions (A1–A6) are determined to be the unrelated pattern (R1–R5) by mistake, which leads to no response of the control system. However, for the model trained with the CNN-AE-S, there are some misclassifications of target movements determined to other target movements by mistake, which leads to abnormal movement of the control system because of the wrong recognition.

To further compare the performance of the proposed methods, Table 1 reveals the mean recognition accuracies and rejection performance for each movement when the recall is defined as 0.8, 0.85, and 0.9. As shown in Table 1, although the traditional machine learning methods of SVM and LDA achieve the highest recognition performance (96.9% and 86.1%, respectively) for target motions, they cannot reject any unrelated samples and they finally misclassify them as target motion tasks. However, other methods are all capable of rejecting unrelated samples depending on the recall factor on the verification dataset. These rejections also affects the ability of the model to recognize the target samples. When the recall factor is set to 0.8, that is, the model has a poor ability to recognize target action, the recognition accuracies of target motion obtained by the LDA-MD, CNN-AE-S, CNN-AE-SC all drop to about 80%, whereas the rejection performance of unrelated motion are 84.0%, 84.3%, 99.7% by these methods, respectively. It is worth noting that the rejection performance of unrelated motion by the CNN-AE-SC is obviously better than the other two methods. Then, when the recall factor is increased to 0.85, the recognition performance of all methods is correspondingly improved to about 85%, whereas the average rejection performance of unrelated movements obtained by the LDA-MD (72.7%), CNN-AE-S (79.3%), and CNN-AE-SC (98.3%) methods slightly drop. The rejection performance of unrelated motion by the CNN-AE-SC method is obviously better than the other two methods. Finally, when the recall factor is defined as 0.9, the recognition performance of all methods for the target movements are correspondingly improved to about 90%, whereas the average rejection performance of unrelated movements by the LDA-MD (59.0%), CNN-AE-S (71.7%), CNN-AE-SC (95.0%) methods drop sharply. it can be seen that with the increase of the recall factor, the proposed method could always maintain better rejection ability for unrelated movements. Therefore, the proposed method is available, and its performance is significantly better than the other two methods, which could be applied to the real-time control of the upper limb exoskeleton tele-rehabilitation robot system. However, all of the above processing flows are performed offline, but how to apply the research framework in this study to the online stage to control the exoskeleton device is a consideration in our future work.

The number of samples in different reconstruction error intervals are shown in Figure 11a,b. The reconstruction loss distribution is achieved by the CNN-AE-S method in the verification dataset and unrelated samples. Compared with Figure 8a,b, although some target movement sample reconstruction errors and unrelated movement sample reconstruction errors can be correctly distinguished, many sample reconstruction errors are wrongly distinguished, indicating that samples are wrongly judged. When the recall coefficient is defined as 0.8, that is, the discrimination threshold is small (threshold = 0.0170, λ=0.00025), the average recognition accuracy of target samples by the CNN-AE-S method and CNN-AE-SC method drops to about 80%. The average accuracy of rejection of unrelated samples by CNN-AE-S methods is 84.3% with an average F-score of 83.7%. The average rejection accuracy is 99.7% for unrelated samples by the CNN-AE-SC method proposed in this paper, with an average F-score of 92.8%. When the recall is defined as 0.9, that is, the discrimination threshold is large (threshold = 0.0235, λ=0.00025). The average recognition accuracy of the two methods for target samples correspondingly increases to about 90%, but the rejection performance of the CNN-AE-S method for unrelated samples drops sharply to 71.7% with an average F-score of 83.8%. The proposed method still has a high rejection performance with an average accuracy of 95.0% and an average F-score of 93.7%. Therefore, the CNN-AE-SC method is superior to the CNN-AE-S method. Although the recognition accuracy of the model for target samples is improved with the increase in recall rate, the rejection performance for unrelated samples will decrease accordingly. However, the model trained with the proposed method has a higher recognition accuracy (90.0%) for target samples and a higher rejection performance (95.0%) for unrelated samples.

The sEMG signal is very sensitive to muscle contraction, and small unrelated movements can introduce interference to the sEMG control system. The unrelated movements will cause the system to execute wrong instructions, which may cause safety accidents while reducing user experience. As shown in Table 2, some comparison results of different methods are summarized. In summary, the existing solutions could be divided into domain-based methods [26] and refactoring-based methods [28], etc.. However, the above work of rejecting irrelevant actions often assumes that the irrelevant actions and target actions are significantly different, so a relatively simple method can be used to distinguish them. Q. Ding [26] constructed an adaptive incremental hybrid classifier by combining one-class support vector data description and multi-class LDA with an average target motions recognition rate of 91.7% and unrelated motions rejection rate of about 90.0%. However, in actual application situations, unrelated actions are unpredictable and often very similar to target actions. Y. Zhang [28], based on a high-density EMG signal, proposed a myoelectric control method to alleviate the interference brought by novel motion tasks with an average target motions recognition rate of 88.6% and the unrelated motions rejection rate of 96.4%. To resolve the above problems, this paper proposes a method for rejecting unrelated motion based on low-density EMG signals that can be used for portable rehabilitation of the wrist joint at home. The results show that the average recognition rate of the method proposed in this paper for the target motion is about 90.0%, and the rejection rate for irrelevant motion can reach 95.0%. Compared with [26] and [28], this method realizes the unrelated motion rejection based on low-density sEMG signals when the unrelated motion and target motion signals are very similar.

## 4. Conclusions

The performance of rejecting unrelated samples is very important for the sEMG-controlled exoskeleton rehabilitation robot. In this paper, a sEMG-based pattern recognition method for wrist rehabilitation is proposed to improve the anti-interference performance of unrelated wrist movements. First, the method uses the idea of metric learning to extract deep distinguishing features from low-density sEMG feature images. Then the automatic encoder is used to calculate the reconstruction error to distinguish the unrelated samples and target samples. Compared with only using softmax loss to train the model, the proposed method enables the model to have a higher recognition accuracy (90.0%) for target samples with a higher rejection performance (95.0%) for unrelated samples, which shows that the proposed method is a viable method to improve the robustness of sEMG-based pattern recognition for exoskeleton rehabilitation robots. In the future, we will further consider the influence of the inter-subject difference of sEMG to model performance and apply it to the real-time control of the upper limb exoskeleton rehabilitation robot system.

## Figures and Tables

**Figure 1 micromachines-14-00555-f001:**
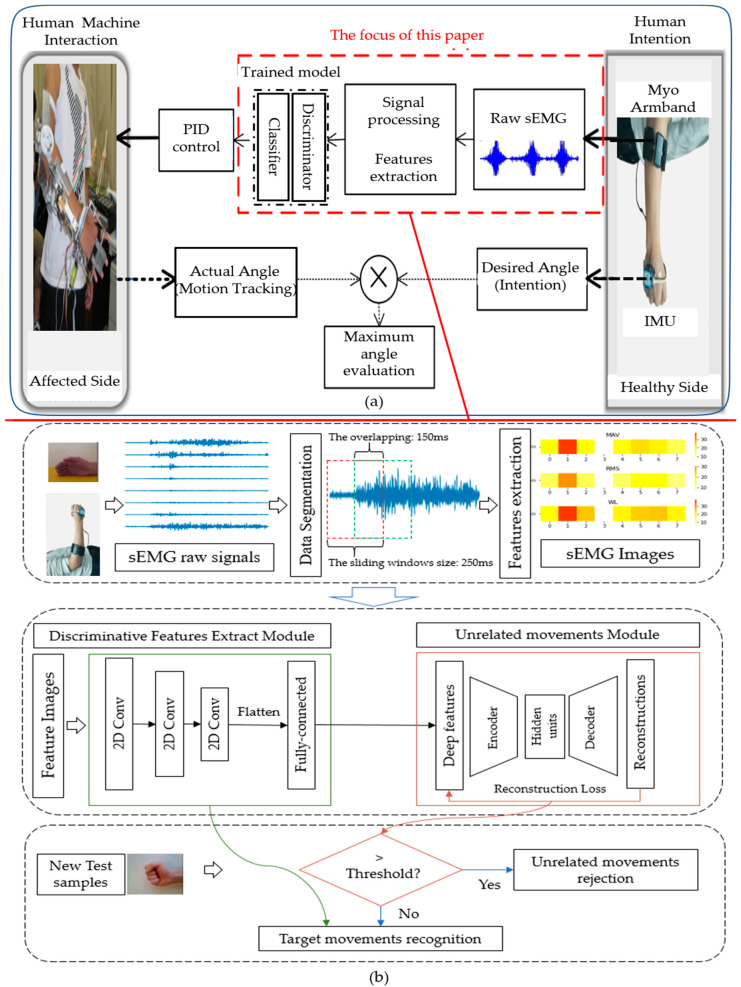
The overview of the system: (**a**) Bilateral rehabilitation based on joint pattern recognition from sEMG signals [4]; (**b**) The overview of the proposed method in this paper.

**Figure 2 micromachines-14-00555-f002:**
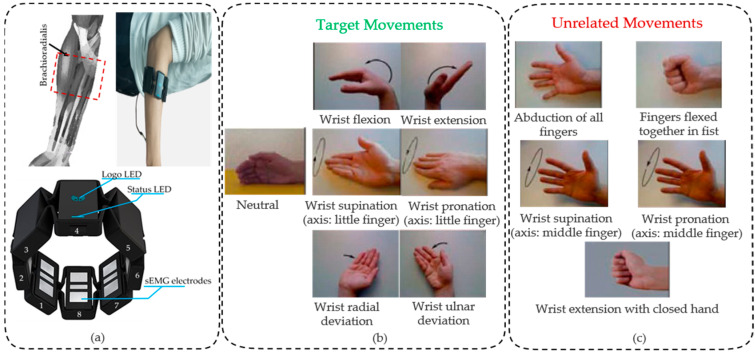
Experimental devices and data acquisition scheme: (**a**) Thalmic Myo armband and schematic diagram of the experimental data acquisition; (**b**) The seven target movements; (**c**) The five unrelated movements.

**Figure 3 micromachines-14-00555-f003:**
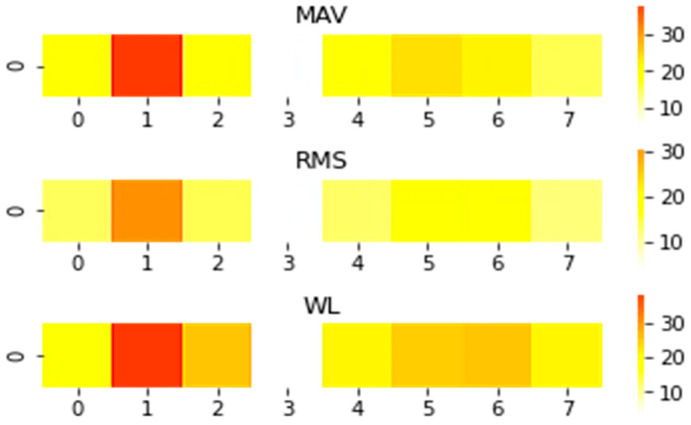
The sEMG images correspond to the target movements of 8 channels’ sEMG signals.

**Figure 4 micromachines-14-00555-f004:**
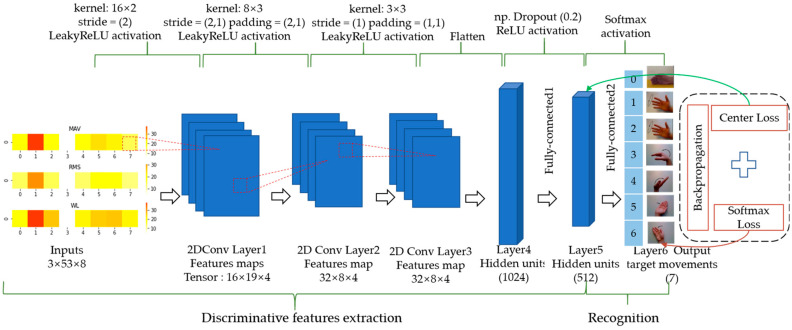
The architecture of the CNN model developed in this paper.

**Figure 5 micromachines-14-00555-f005:**
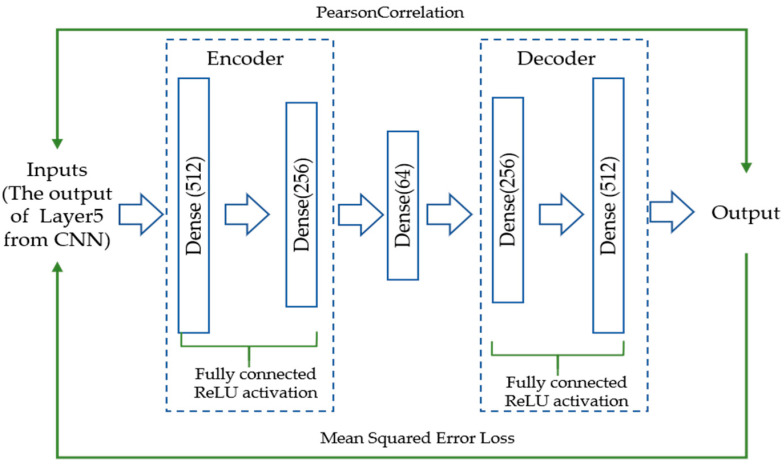
The architecture of the discriminant network using multiple autoencoders.

**Figure 6 micromachines-14-00555-f006:**
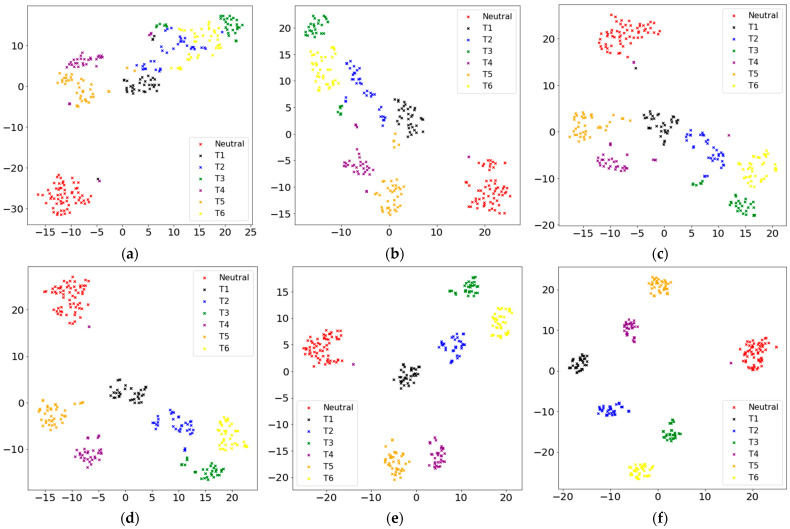
Visualization of deep features extracted from all subjects and the scattered points represent samples with colors to indicate different motions: (**a**–**i**) Visualization of deep features when λ is set as 0, $5\times 10^{-6}$, $1\times 10^{-5}$, $1.5\times 10^{-5}$, $2\times 10^{-5}$, $1\times 10^{-4}$, $2.5\times 10^{-4}$, $5.0\times 10^{-4}$, and $1\times 10^{-3}$ respectively.

**Figure 7 micromachines-14-00555-f007:**
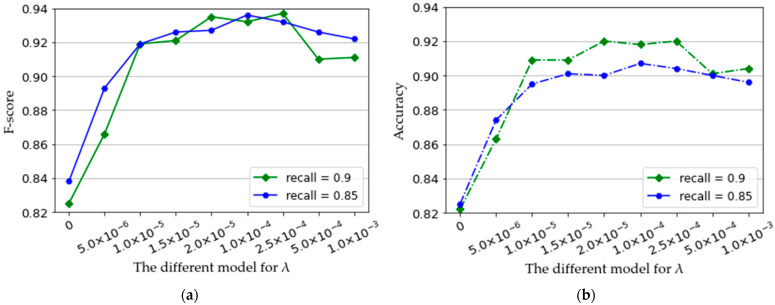
The mean F-score (**a**) and mean accuracy (**b**) of all subjects when the recall is defined as 0.85 and 0.9, respectively, achieved by the trained models with different λ.

**Figure 8 micromachines-14-00555-f008:**
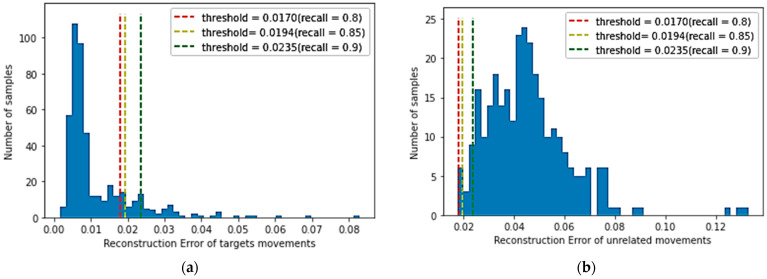
Reconstruction loss distribution achieved by the trained model: (**a**) Reconstruction loss distribution on the validation dataset; (**b**) Reconstruction loss distribution of all unrelated movements; (**c**) Pearson correlation(PC) distribution for each movement on the test dataset and all unrelated samples when the recall is 0.85; (**d**) Pearson correlation(PC) distribution for each movement of test dataset and all unrelated samples when the recall is 0.9.

**Figure 9 micromachines-14-00555-f009:**
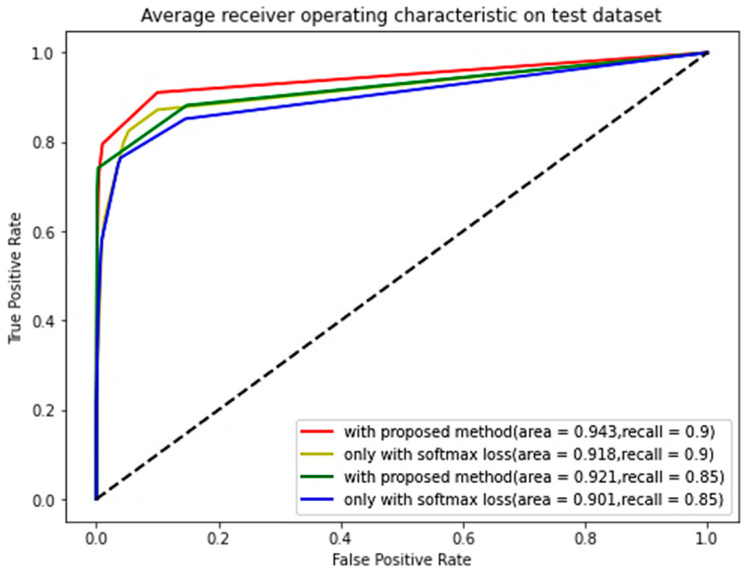
ROC curves are described by the CNN-AE with softmax loss and center loss and CNN-AE with softmax loss when the recall is 0.85 and 0.9.

**Figure 10 micromachines-14-00555-f010:**
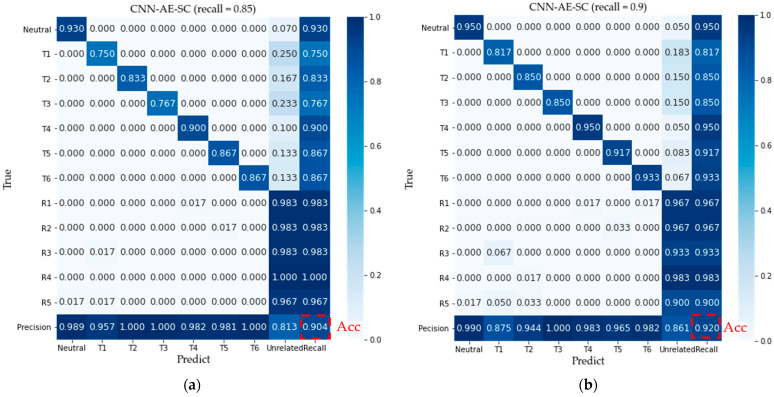
Confusion matrices averaged over all subjects: (**a**,**b**) Confusion matrices for the proposed method on the test dataset and all unrelated samples when the recall is defined as 0.85 and 0.9 (**b**); (**c**,**d**) Confusion matrices for CNN-AE-S method on the test dataset and all unrelated samples when the recall is defined as 0.85 and 0.9; Neutral and T1–T6 denote seven target movements, whereas R1–R5 denote rejection movements.

**Figure 11 micromachines-14-00555-f011:**
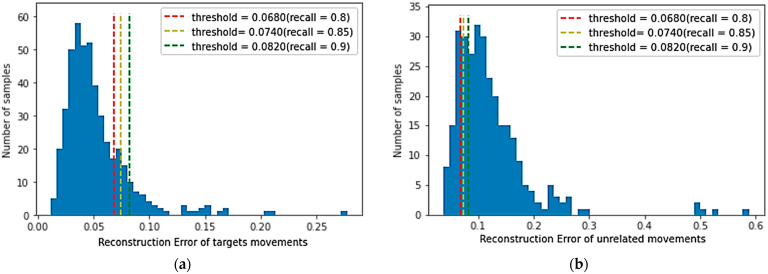
Reconstruction loss distribution achieved by the CNN-AE-S. (**a**) Reconstruction loss distribution in the validation dataset; (**b**) Reconstruction loss distribution of all unrelated movements.

**Table 1 micromachines-14-00555-t001:** Recognition accuracies of target and unrelated movements and F-score when the recall factor is set to 0.8, 0.85, and 0.9, respectively.

Recall	Method	Target Movements	Unrelated Movements						
			Averaged	R1	R2	R3	R4	R5	
		Acc (%)	(%)	Rejection Performance (%)					Averaged F-Score (%)
	SVM	96.9	-	-	-	-	-	-	-
	LDA	86.1	-	-	-	-	-	-	-
0.8	LDA-MD	77.4	84.0	68.3	96.7	85.0	76.7	93.3	78.5
	CNN-AE-S	80.4	84.3	88.3	96.7	86.7	78.3	71.7	83.7
	CNN-AE-SC	82.4	99.7	100	100	100	100	98.3	92.8
0.85	LDA-MD	82.2	72.7	56.7	95.0	65.0	56.7	90.0	75.5
	CNN-AE-S	85.4	79.3	85.0	93.3	81.7	68.3	68.3	82.5
	CNN-AE-SC	85.2	98.3	98.3	98.3	98.3	100	96.7	93.2
0.9	LDA -MD	85.4	59.0	45.0	90.0	45.0	33.3	81.7	72.2
	CNN-AE-S	89.8	71.7	73.3	88.3	76.7	63.3	56.7	83.8
	CNN-AE-SC	90.0	95.0	96.7	96.7	93.3	98.3	90.0	93.7

**Table 2 micromachines-14-00555-t002:** The comparison with the related studies.

	sEMG Channels	sENG Acquisition	Performance	
			Target Motions Recognition (%)	Unrelated Motions Rejection (%)
Q. Ding [26]	5	Delsys, Trigno	91.7	90.0 ± 5
Y. Zhang [28]	6 × 8	High-density sEMG	88.6	96.4
Z. Bi [32]	6 × 8	Wearable EMG Bridge	94.0	-
This work (recall = 0.9)	8	Thalmic Myo armband	90.0	95.0

## Data Availability

Not applicable.

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
