# Peer review of "Low-Density sEMG-Based Pattern Recognition of Unrelated Movements Rejection for Wrist Joint Rehabilitation"

_micromachines, 2023, doi:10.3390/mi14030555_

Round 1

Reviewer 1 Report

Authors have carried out nice piece of work on "Low-density sEMG-based Pattern Recognition of Unrelated 2 Movements Rejection for Wrist Joint Rehabilitation”. The manuscript is extremely well written, easy to comprehend. Authors have explored the aspects to improve the anti-interference performance of wrist unrelated movements. The information provided in this manuscript useful to the readers, adds a value to the scientific community. The manuscript can be acceptable for publication in its current form after addressing few minor points.

1.      Authors would need to improve the clarity of Figure 1.

2.      The text in the figure 2 is not readable, please increase the text size, same with the figure 4.

Reviewer 2 Report

The work presents a method of classification of wrist sEMG signals to identify gestures unrelated to wrist joint rehabilitation with the transformation of the sEMG signal to image and processing by a CNN. The proposal of the work is interesting, but a lot of information and some methodological concepts are missing.

Some phrases should be revised to improve the quality of paper. Many sentences are starting with a capital letter in the middle of the sentences, as in lines 110, 130 and 140. Figures are presented before their mention on text. Figure should start with capital letter on text.

Abstract

Remove the explanation about the acronyms.

Introduction

Lines 39-40. “In addition, sEMG signals are thought 300ms advanced occur than the movement of the limb”. Please, confirm this information because the cited paper mention another work.

Line 65: Reference cited is not [6], but reference [26]. Please, review all the cited works.

Methods

Figure 1 is presented before than their citation on text. Improve the quality of Figure 1. Use a legend as (a), (b)…(n) to explain this details. It is more useful to describe on section 2.1 the system behavior.

Line 108: Reference the study.

Why was used a CNN? Couldn't simpler classifiers like SVM and LDA give similar results? Why were these models not tested, as they are the gold standard for analyzes involving sEMG?

The action of the five rejection movements was not clear in the methodology. Please, make it more explicit.

What are the features of the implemented 50Hz notch filter? What software was used for its implementation?

Improve the quality of Figure 2. The caption should follow the journal’s template.

Was not sEMG signal identification method applied, such as the double onset?

What is the difference between transforming sEMG signals into images and using them numerically in a CNN? This is not clear in the text and it is not shown by the authors whether there is any advantage.

Why were RMS, MAV and WL chosen as features to be extracted? There are other features that are more robust for sEMG classification systems, such as L-Scale and Maximum Fractal Length. Why were these not considered?

One of the characteristics of CNNs is the possibility of not necessarily needing feature extraction steps. Because this path was not considered to reduce the number of processing steps, which tend to increase the computational cost.

Line 202: “Many researches”. Authors did not cite any research.

How did the authors arrive at the CNN topology?

Lines 225-226: Reference this sentence.

The references of equations 6-12 are missing.

Experiments results and discussion

Figure 7: Label the axis of each graph. Follow the journal’s template on caption.

Improve the quality of Figure 8.

Figure 8. What could be related to the peak increase in the f-score value with λ in places where there is not the greatest separability of classes, since the greater λ, the greater their separability.

Figure 9 and 10: Improve the quality.

Remove the black cells on Figure 11 to increase the visibility.

What is the reason for the drop in accuracy at T1 and T2 for items (c) and (d) in Figure 11?

Figure 11: Improve the quality and insert the items (c) and (d) on caption.

The authors did not develop a comparison of their work with related studies in the discussion. This point is very important so that the authors can highlight their contribution.

Conclusion

Future works are missing.

References

Some references are not in template. Please, correct them.

Reviewer 3 Report

This paper proposes a method that incorporates the center loss into the structure of a convolutional neural network (CNN) with SoftMax loss to improve the rejection of five unwanted hand gestures from seven target hand gestures based on sEMG signals acquired on normal subjects using an armband and which is intended to be used for wrist joint rehabilitation purposes. The proposed method improves the rejection accuracy.

Because the intended application of the method is to be part of a wrist joint rehabilitation process, the authors should include information supporting the plausibility of the pool of target gestures and unrelated gestures for this objective, highlighting the unpredictability and similarity they mentioned (line 74). Otherwise, the mention of the future application “for Wrist Joint Rehabilitation” within the title should be omitted not to mislead the reader.

The models, which achieved a high recognition accuracy and rejection performance, were trained and tested using data from two randomly selected datasets, the first including the data of 10 participants and the second one the data of 2 participants. It is essential to state specific details about the randomization process followed by the authors to build their training, validation, and test datasets, if they either randomized the subjects or just the samples considering all subjects (line 165). If authors did not randomize subjects, this could increase artificially because there are very close samples in the training dataset, validation dataset, and test dataset due to the nature of the sliding window method used to process the signals. In that case, it is advisable to run the process using subjects as randomization units to check the performance of their plans.

The authors must clarify how the Myo armband sensors distribution shown in Figure 2(a) is assigned, considering muscle group distribution is not homogeneous while Myo armband sensors are. Also, the authors should clarify if they are not considering muscle groups other than the ones shown in figure 2(a), which are expected to be sensed by the Myo armband sensor at the back of the forearm.

The authors should describe the reasons which motivated the decision about the architecture of the CNN shown in figure 4.

The authors stated that the model's performance increases as lambda increases; however, figure 8 suggests a decrement in the model's performance when lambda grows.

More discussion is needed to compare the authors' results to other classifiers made by other authors. The proposed method reaches the same accuracy as previous ones, improving the rejection performance. Authors should discuss the pros and cons of the proposed method considering its complexity and applications.

 It is suggested to check the full text to identify and correct grammatical errors, mistakes, and omissions. Here are some examples:

1.      Definition of abbreviations on their first apparition: Line 89.

2.      Misspelling of words: line 110.

3.      Matching of personal pronouns: line 122.

4.      It is necessary to define the meaning of “neutral” in line 137.

5.      Punctuation mistakes: line 140.

Reviewer 4 Report

The paper proposes rejection technique of unrelated movements to improve rehabiliation system of wrist joint using sEMG sensors. The text is well written and detailed the models, CNN architecture, results and discussion. However, the paper must be reviewed according to the following comments:

- Correct some missing words and expressions: figure 1, actrual angle -->actual angle; ...

- Eq 1: define the two right terms.

- There is a confusion between the following words: static state, resting state, static signal, window signal.

- Eq 2: the notation 'n' is the total number of samples for each sEMG channel or only resting state?

- Eq 3: 'n' is used as index while it's denoted before as total number of samples?

- Eq 2: it's not used in the work nor feature nor in other expression?

- Eq 12: review the expression of the denominator ( standard deviation).

- it's recommended to compare the performances with other NN architectures and/or other published techniques?

- The sEMG band frequency is limited to 100Hz , as the sampling rate is 200Hz. Further information and consequently features  are in the standard band up to 500 Hz. 

* What's the impact of the sampling frequency on the performances?

* What's the impact of the sliding window size?

* The hyper parameter 'lambda' should be detailed.

Round 2

Reviewer 2 Report

The authors did the requested reviews. I consider that the paper have quality for publishing now.